# Library Preparation Based on Transposase Assisted RNA/DNA Hybrid Co-Tagmentation for Next-Generation Sequencing of Human Noroviruses

**DOI:** 10.3390/v13010065

**Published:** 2021-01-06

**Authors:** Zilei Zhang, Danlei Liu, Dapeng Wang, Qingping Wu

**Affiliations:** 1Department of Food Science and Technology, School of Agriculture and Biology, Shanghai Jiao Tong University, Shanghai 200240, China; allyzhangzl@sjtu.edu.cn (Z.Z.); liudanlei@sjtu.edu.cn (D.L.); 2State Key Laboratory of Applied Microbiology Southern China, Guangdong Provincial Key Laboratory of Microbial Culture Collection and Application, Guangdong Open Laboratory of Applied Microbiology, Guangdong Institute of Microbiology, Guangzhou 510030, China

**Keywords:** human norovirus, next generation sequencing, library preparation, Tn5 transposase, TRACE-seq

## Abstract

Human noroviruses (HuNoVs) are one of the leading causes of foodborne illnesses globally. The viral genome is the most essential information for viral source tracing and viral transmission pattern monitoring. However, whole genome sequencing of HuNoVs is still challenging due to the sequence heterogeneity among different genotypes and low titer in samples. To address this need, in this study, the Transposase assisted RNA/DNA hybrid Co-tagmentation (TRACE-seq) method was established for next generation sequencing library preparation of HuNoVs. Our data demonstrated that almost the whole HuNoVs genome (>7 kb) could be obtained from all of the 11 clinical samples tested. Twelve genotypes including GI.3, GI.4, GI.5, GI.8, GII.2, GII.3, GII.4, GII.6, GII.12, GII.13, GII.14, and GII.21 were involved. Compared with the traditional method for viral metagenomics library preparation, optimized TRACE-seq greatly reduced the interference from the host’s and bacterial RNAs. In addition, viral genome sequences can be assembled by using less raw data with sufficient depth along the whole genome. Therefore, for the high versatility and reliability, this method is promising for whole viral genome attainment. It is particularly applicable for the viruses with a low titer that are mixed with a complicated host background and are unable to be cultured in vitro, like the HuNoVs utilized in this study.

## 1. Introduction

Human noroviruses (HuNoVs) are the leading cause of non-bacterial epidemic gastroenteritis globally and cause an estimated 684 million illnesses of acute gastroenteritis worldwide, resulting in approximately $60 billion in societal costs [1]. A norovirus (NoV) has a single-stranded, positive-sense genomic RNA of approximately 7.5 kb, that have a 5′-terminal virus protein, are genome-linked (VPg), and possess 3′-terminal poly (A) [2]. Based on phylogenetic analysis of major capsid protein (VP1) amino acid sequences, at least 10 norovirus genogroups (GI-GX) and 48 genotypes have been recognized [[3],]. The uncorrected pairwise distance ranges for strain, genotype, and genogroup were 0–14.1%, 14.3–43.8%, and 44.9–61.4%, respectively [4]. No routine cultivation method in vitro was available for different genotypes of HuNoVs isolation and replication [5]. To prevent and control the HuNoVs outbreaks, the key is to identify the viral pollution source and monitor viral transmission patterns based on the viral genome sequences [6].

Metagenomic next-generation sequencing (NGS) is an effective approach for broad-spectrum pathogen identification in clinical samples, as nearly all potential pathogens (viruses, bacteria, fungi, and parasites) can be detected by identified unique DNA and/or RNA sequences [7]. Viral genome sequencing is a branch of RNA sequencing (RNA-seq), and relies on second strand complementary DNA (cDNA) synthesis to generate initial material for library preparation. The common protocols contain a few key steps, including RNA extraction, poly-A selection or ribosomal RNA depletion, reverse transcription, second strand complementary DNA (cDNA) synthesis, adapter ligation, and PCR amplification [8,9,10]. Viral metagenomics can be realized in two ways: one is to enrich viral RNA by the poly A-capture method to remove as many hosts and bacterial rRNAs as possible (mRNA-seq), the other is to use ribosomal random primers for background sequences deletion, i.e., primers selected among random hexamers that match rRNA genes from the host or the matrix [11,12].

Traditional methods still have limited capacity to remove background data for viral genome sequencing, though host or background nucleic acids can account for up to 99% of the raw reads [12]. A crucial part of NGS is to increase the minimal required number of sequencing reads per sample (SRPS) in order to get a reliable in-depth analysis. Underestimation of SRPS results in a low yield of target sequences. The known minimum required number of SRPS was 0.008% for the Epstein-Barr virus (EBV) [13] and 0.3% for the Zika virus [14]. The existence of background data increases the requirements for sequencing data and the cost of each sample. Meanwhile, it also increases the difficulty of subsequent bioinformatics analysis, sample analysis times, and data storage costs. In addition, it is difficult to design primers that work for a broad range of strains with high sensitivity and efficiency, because of sequence heterogeneity among HuNoVs genomes [15]. Therefore, it is a big challenge to obtain whole genome sequences for HuNoVs compared with other viruses due to their sequence heterogeneity and complicated background data.

Tn5-mediated transposition of double-strand DNA has been widely utilized in various NGS tagmentation. DNA sequencing library preparation based on Tn5 transposase could be processed conveniently and has low requirements for sample inputs [16,17]. Tn5 transposase has been altered to be capable of direct tagmentation of RNA/DNA hybrids in vitro in 2020, named Transposase-assisted RNA/DNA hybrids Co-tagmEntation (‘TRACE-seq’) [18,19]. TRACE-seq offers a greatly simplified protocol and produces results with higher reproducibility and GC uniformity compared with prevailing RNA-seq methods [18]. However, this method is still at the theoretical stage and has not been applied to clinical samples. In this study, HuNoVs clinical samples were used to evaluate the strategy for establishing a library preparation method for specific viruses based on the theory of TRACE-seq. Each sample was divided into tube A and B for reverse transcription using the TRACE-seq method, which optimized the method of library preparation for NGS of HuNoVs (Figure 1). The RNA/DNA hybrid was tagmented by the modified Tn5 transposase and partial sequencing adaptors were added to fragment ends simultaneously. By achieving specific recognition of HuNoVs viral genomes, the percentage of effective reads could be increased, and the difficulty of complete reverse transcription caused by high GC regions can be overcome.

## 2. Materials and Methods

### 2.1. RNA Extraction and RT-qPCR Identification of Clinical Samples

HuNoVs-positive clinical samples were kindly provided by Dr. Zhiyong Gao at the Beijing Center for Disease Control and Prevention (CDC), Dr. Ningbo Liao at the Zhejiang CDC, and Dr. Huiying Li at the Chinese CDC [20]. Viral RNAs were extracted from 140 μL of 10% stool suspension with a HiPure Viral RNA Kit (Magen, China) according to the manufacturer’s instructions without carrier RNA. RNA was eluted with 30 μL RNase-free water. Duplicate one-step RT-qPCR assays for GI and GII HuNoVs were detected in a single tube using a Biorad CFX96 qPCR machine [20,21].

### 2.2. Primer Sets Design

The complete sequences of GI and GII HuNoVs before Oct 2019 (*n* = 1137) were downloaded from the National Center for Biotechnology Information (NCBI, https://www.ncbi.nlm.nih.gov/, accessed on Oct 2019) and aligned using mafft to identify conserved regions for primers design separately [22]. Principles for reverse transcription primers selection including (1) high conservation of the fragment region, (2) length with 12–14 bp, and (3) interval with 1000–1500 bp. Primers designed in this study are listed in Appendix A. Primers were synthesized by Sangon Biotech Co., Ltd. (Shanghai, China) and diluted to 200 μM as stock-solutions. They were further configured into primer sets with a final concentration of 10 μM according to the requirements of tube A or B.

### 2.3. TRACE-Seq Library Preparation and Sequencing

Libraries were prepared using TruePrep^®^ RNA Library Prep Kit for Illumina (Vazyme Biotech Co., Ltd., Nanjing, China) with some modification. RNA (8.0 μL) was heated at 65 °C for 5 min and quickly placed on ice for 2 min. It was then mixed with 2.0 μL 5 × genomic DNA wiper Mix and incubated at 42 °C for 2 min to remove genomic DNA. The solution was then divided into tube A / tube B equally and added to the different primer sets described in Section 2.2. Then 1.0 μL 10 × RT Mix, 0.5 μL HiScript III Enzyme Mix, 1.0 μL Primer sets, and 2.5 μL Nuclease-free ddH_2_O were added into each tube for reverse transcription. Reaction conditions were as follow: 25 °C and 5 min, 37 °C and 30 min, and 85 °C and 5 s. Transcription products in tube A and tube B were combined, then we added 10.0 μL tagment buffer and 2.0 μL Tn5 VR1, then pipetted 20 times gently to mix well. The tagmentation conditions were 55 °C and 15 min and the solution was then held held at 10 °C. After the reaction was completed, 2.0 μL TStop Solution was immediately added to the reaction solution. After using a vortex for mixing, the solution was then incubated at room temperature for 5 min. Then we added 50 μL VAHTS HiFi Amplification Mix, 5.0 μL PCR Primer Mix 3 for Illumina, and 1.0 μL TSE into the solution (34 μL) and incubated it at 55 °C for 5 min, 60 °C for 5 min, and 95 °C for 5 min for extension. TruePrep^®^ Index Kit V2 for Illumina (Vazyme, Nanjing, China) N5XX (5.0 μL) and N7XX (5.0 μL) were added according to the number of samples and index matching strategy. The cycling conditions were as follows: initial denaturation at 95 °C for 3 min, 20 cycles of PCR with 98 °C for 20 s, 60 °C for 15 s, 72 °C for 30 s, and final extension at 72 °C for 5 min. The library was purified using 80 μL (0.8×) VAHTS^®^ DNA Clean Beads (Vazyme, Nanjing, China). Sequencing was performed using the Illumina X-10 platform to generate 2 × 150 bp reads.

### 2.4. Bioinformatics

Reads from the Illumina X-10 were trimmed using fastp to remove sequences with low quality [23]. Clean reads were de novo assembled by SPAdes [24] or MEGAHIT [25]. Bowtie2 was used to extract the sequence of HuNoVs before assembling when the result of a de novo sequence was not ideal [26]. Contigs were identified using BLASTn based on a curated reference list that consists of all HuNoVs complete-genome sequences in GenBank that were uploaded before Oct 2019 (*n* = 1137) [27]. Sequences were then validated and genotyped using a web-based genotyping tool (http://www.rivm.nl/mpf/norovirus/typingtool, accessed on Dec 2020, Version2.0) [28,29]. Mapped reads to contigs (CLC genome workbench) was used to identify coverage reads of HuNoVs per sample. After mapping reads to the foodborne pathogen database, sequence taxonomy was assigned by Kraken2 [30,31]. Data unification and heat map drawing were realized by using TBtools [32]. The raw sequence data reported in this paper have been deposited in the Genome Sequence Archive [33] in National Genomics Data Center [34], Beijing Institute of Genomics (China National Center for Bioinformation), Chinese Academy of Sciences, under the accession number CRA003583 that is publicly accessible at https://bigd.big.ac.cn/gsa, accessed on 4 Dec 2020.

## 3. Results

### 3.1. Sequencing Results

Eleven HuNoVs positive clinical samples were used for method development (Figure 2). Raw data for each sample was 6–240 Mb. A HuNoVs genome above 7 kb could be obtained for all tested samples. The BLAST score is a number used to assess the biological relevance of a sequence. All the sequences were above 70.0, indicating that the assembled sequences are of high similarity with the norovirus sequences in the database. BLAST scores of sequences belonging to the rarely reported genotypes including GI.3[P10], GI.4[P4], GI.5[P5], GI.8[P8] were relatively low. This situation was mainly caused by the lack of sufficient co-responding genotyped whole genome sequences in the database. Most of the sequences cover the entire three ORF regions, while the 5′ or 3′ end sequence was missing in a few samples, which has no effect on the genotyping or evolution analyzing of the main coding protein region. Genotypes involved in this study including GI.3, GI.4, GI.5, GI.8, GII.2, GII.3, GII.4, GII.6, GII.12, GII.13, GII.14, and GII.21, indicated that this method is suitable for a wide range of HuNoVs genotypes. Samples SJTU-1030 and SJTU-7565 were found to contain two different genotypes of HuNoVs strains, respectively. Reads coverage were listed in Figure 2. No special location was found with insufficient sequencing depth.

Three different genotypes of HuNoVs were identified in sample SJTU18C7, while only one full-length viral genome was obtained. A full-length genome of GII.13 and two genome fragments with 3 kb of GI.8 and GII.21 were assembled. According to our previous study, one strain was always found to be dominant in multiple infections in one patient with more than 65% mapped reads [35]. This situation might lead to underestimation of SRPS for the minor strains in the multiple infection sample, thus whole genomes of all infected strains are unable to be obtained. Since this method is highly targeted and the amount of raw data for different samples varies greatly, it is recommended that the library should be quantified by qPCR library quantification to effectively evaluate the sequencing volume distribution of samples before loading the sample in the chip.

### 3.2. Effective Data Distribution

In the traditional method for viral metagenomic, mRNA in extracted RNA solution is captured for library preparation while the mRNA of host and bacteria are also involved, thus interfering with the analysis of viral genome (Figure 3). Meanwhile in the TRACE-seq method, the proportion of HuNoVs reads strongly increased through the reverse transcription by specific primers, while reads of food-borne pathogenic bacteria are reduced from 10^5^ to 10^2^ (Figure 3). However, other sequence interferences still exist in this method. Fusarium was found to be the main interfering pathogen in tested samples. This may be related to the undesired spurious priming between designed the primers and genome of Fusarium. The situation could be resolved by increasing the amount of sequencing data. It can also be seen from the distribution map that this method is highly pertinent and cannot achieve metagenome analysis of poly-A microorganisms in the same way as the traditional mRNA method, thus this technique lacks information on other pathogenic microorganisms in the sample.

## 4. Discussion

Sequencing of viral genomes has become a pivotal part of virological research [36]. Whole genome or long sequence is essential for either tracing the contaminating event or identifying emerging variants or viruses. Therefore, efforts have been paid towards exploring a method for the specific enrichment of HuNoVs sequences. Sureselect is a target enrichment method using a panel of custom-designed 120-mer RNA baits that are complementary to all publicly available HuNoVs sequences [37]. Metagenomic sequencing with spiked primer enrichment (MSSPE) enriches targeted viral RNA sequences by using spiked primers [38]. Another strategy to enhance the recovery of HuNoVs in complex matrices is the capture-based metagenomics (ViroCap), where nucleic acid libraries are enriched in viral sequences using probes [39]. Although many target enrichment arrays were designed to capture virus sequences, the target sequences still represent a small proportion of the total reads, thereby suggesting non-specific capturing [12]. Therefore, it is necessary to explore library preparation methods aiming for the target virus.

Our method successfully combined the advantages of transposase with the genome characteristics of HuNoVs, and verified that the TRACE-seq theory can be applied to the whole genome sequencing of RNA virus samples. Due to the high genetic diversity of the HuNoVs genus, it is difficult to design primers that work for a broad range of strains with high sensitivity and efficiency [15]. The TRACE-seq method does not need to design PCR primers of about 20 bp, as short-chain reverse transcription primers of only 12–14 bp work efficiently with high specificity. Therefore, it is very suitable for the whole genome sequencing of HuNoVs. In this study, segmented design of the reverse transcription primers was implemented to avoid the degradation of the target viral RNA and the inability of full-length reverse transcription due to the existence of a high GC region. In addition, in order to maximize the coverage of all regions of the genome and avoiding the genome sequencing errors caused by primer addition, two tubes A and B were operated simultaneously in the reverse transcription step with different primer sets.

This method is aimed towards whole genome analysis of a specific virus sequence for HuNoVs, and thus information of other microorganisms in the sample would be missed. It is suitable for samples whose pathogens have been confirmed through nucleic acid or immune assays that have been previous identified but are not suitable for unknown pathogen metagenomics. The method reduces the operating steps of the process, which minimizes the consumption and loss caused by the library preparation process. The magnetic bead purification step only needs to be done after the final PCR amplification, which avoids cDNA second-strand synthesis and ssDNA purification operations, and therefore minimized the loss of target fragments. The cost of the library for each sample is only approximately $50. By excluding a large amount of metagenomic information, and only analyzing specific viruses, the probability of obtaining a whole genome of the target virus is improved, thereby potentially reducing sequencing costs substantially. In addition, since only a small amount of data is needed for subsequent sequencing, more samples could be sequenced on a chip by labeling samples with different indexes.

## 5. Conclusions

The method developed in this study provides a promising path for viral genome acquisition, especially for viruses with a low titer, that are mixed with a complicated host background, and that cannot be cultured in vitro. Obtaining whole genome information is the best preservation and analysis method for these viruses. Optimized TRACE-seq processes provide high versatility and reliability, which is very suitable for obtaining information for a specific viral whole genome.

## Figures and Tables

**Figure 1 viruses-13-00065-f001:**
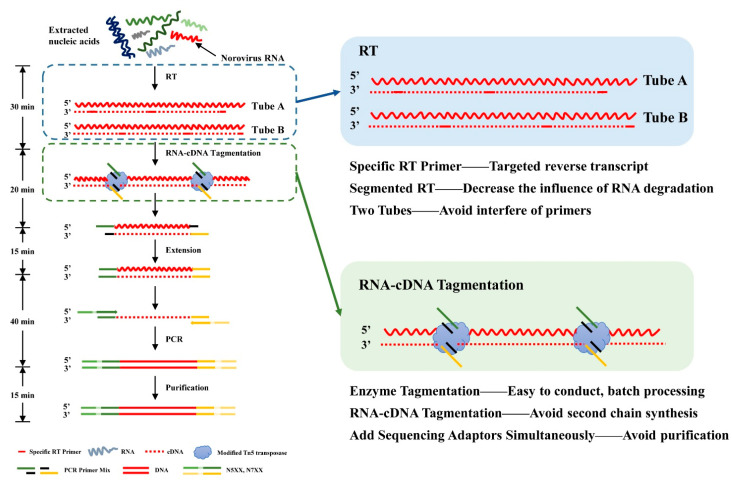
Workflow of the next-generation sequencing (NGS) library preparation strategy based on Transposase-assisted RNA/DNA hybrids Co-tagmEntation (termed TRACE-seq) Library construction processes are listed in the left panel. The main optimizations of this method are marked with two boxes and zoomed to show details with advantages listed in the right panel. The process consists of four components: RNA reverse transcription, RNA/cDNA hybrid tagmentation, PCR amplification, and purification. Specifically, RNA is divided into tube A and B for reverse transcription using primers designed for human noroviruses (HuNoVs). The RNA/DNA hybrid is then tagmented by the modified Tn5 transposase directly. High-fidelity DNA polymerase then amplifies the cDNA into a sequencing library after initial end extension. The purified product is an indexed library which is ready for sequencing.

**Figure 2 viruses-13-00065-f002:**
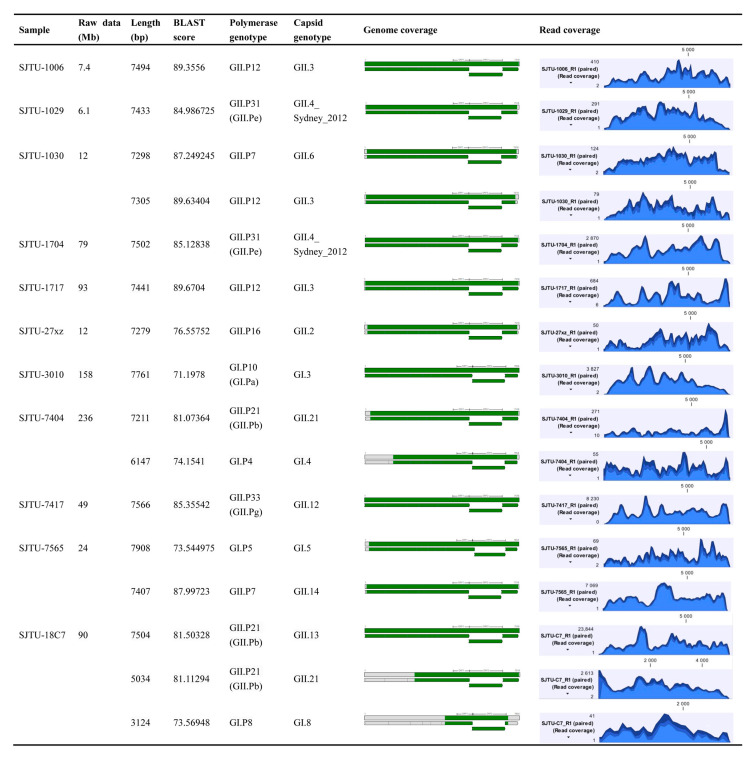
Detailed sequencing results of HuNoVs positive samples using TRACE-seq. Results were analyzed by each sample. Sequences which could be genotyped by polymerase and capsid are listed. In the genome coverage column, sequences obtained in this study that matched the reference sequence of HuNoVs are shown in green. In the last column, altitude indicates the reads coverage in each position of obtained sequences.

**Figure 3 viruses-13-00065-f003:**
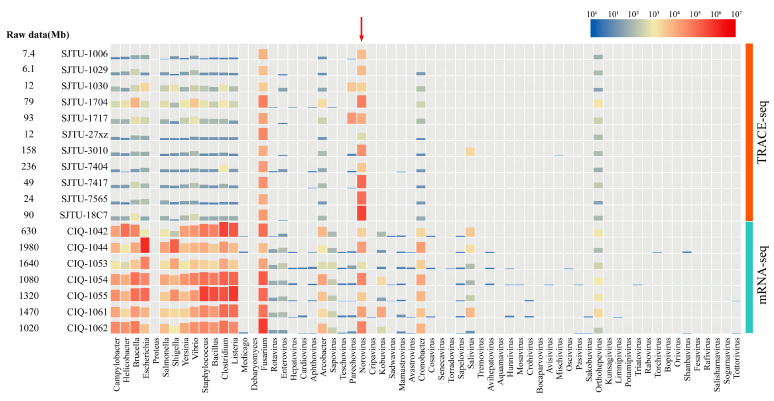
Species distribution heatmap of effective reads of the TRACE-seq method and mRNA-seq method. Samples are listed by the left side, the main food-borne pathogenic microorganisms are listed in abscissa. The effective reads number of each pathogen are shown by the height of column and color from blue to red by the increase of the reads number. The relationship between the effective reads number and the related color is shown on the upper right site. Reads of norovirus are pointed to with a red arrow.

## Data Availability

Data is contained within the article or Appendix A.

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
