# Peer review of "Library Preparation Based on Transposase Assisted RNA/DNA Hybrid Co-Tagmentation for Next-Generation Sequencing of Human Noroviruses"

_viruses, 2021, doi:10.3390/v13010065_

Round 1

Reviewer 1 Report

Due to large sequence heterogeneity across Human Norovirus (HuNoV), there is a need for an effective sequencing technique that can accurately distinguish between different HuNoV serotypes in clinical samples. Under certain conditions, the commonly used RNA-seq method can be prone to contamination from the host RNA, may be expensive and can lead to low HuNoV sequence coverage. Thus, an improved method for next generation sequencing of clinical norovirus isolates would be beneficial.

This manuscript by Zhang and colleagues describes a method for improved HuNoV sequencing using the previously developed method (e.g Picelli et al. 2014) of transposase-assisted RNA/DNA hybrids Co-tagmentation based sequencing (TRACE-seq). The author demonstrate that TRACE-seq is able to detect a large range of HuNoV serotypes, including those most clinically relevant (such as GII.4), with good genome coverage for most of the genotypes identified. Furthermore, the authors data suggest that TRACE-seq can result in sequences with less contamination from other micro-organisms compared to different samples analysed by RNA-seq. Overall this paper indicates TRACE-seq could be a useful tool in detecting HuNoV serotypes in clinical samples. The manuscript is generally well written, but can be editorially improved in places. The work appears to be well conducted and the results are novel. However, there is some minor work that should be done to paper in my opinion, specifically:

Major Comments:

  • The main experimental concerns are that the TRACE-seq results have not been compared effectively to the RNA-seq method that this paper states it supersedes. It is explained in the introduction that RNA-seq gives poor yields and so the paper needs to demonstrate this in the data. For example, I would suggest analyzing the same sample both via RNA-seq and TRACE-seq in Figure 2 , so a direct comparison can be conveyed between the BLAST score, serotypes identified, genome coverage and read coverage. This is the best way to accurately determine whether TRACE-seq is a more effective sequencing method. Perhaps the authors have done this but the data isn’t presented effectively enough.
  • Figure 3 attempts to compare between TRACE-seq and RNA-seq more successfully that Figure 2. However, these are still not direct comparisons, as the samples analyzed by TRACE-seq are different to the samples analyzed by RNA-seq (i.e. SJTU-1006 and CIQ-1042). The results show that TRACE-seq samples have less contamination than the RNA-seq, but if there is no direct comparison then it can’t be expressed that this is only due to the type of sequencing technique applied. I would again analyze the same sample by TRACE-seq and RNA-seq and compare the contaminants present.

Minor Comments:

  • There is a lot of data in Figure 2 fit into a relatively small space and so the text is very small – especially on the genome coverage section. I can’t read what it says even when I zoom in on my PDF reader software. If the data was spread across two pages this would be displayed more effectively.
  • For completeness, what the BLAST score means could be briefly explained in the text, so it’s importance is communicated to the reader.
  • One of the techniques is referred to as ‘RNA-seq’ throughout the text but in Figure 3 the technique is referred to as ‘mRNA-seq’. Change to make consistent.
  • Although the English is generally very good, there are a few grammatical errors that should be changed. A few examples:
    • Line 61: ‘The known minimal required number’ – change to ‘minimum’
    • Line 64-65: ‘It is difficulty to design primers’ – change to ‘difficult’
    • Line 181: ‘Situation could be solved….’ – change to ‘The situation could be solved’
    • Line 184-185: ‘thus lacks information on other pathogenic microorganisms in the sample’ – change to ‘thus it lacks’ or ‘thus this technique lacks’

Reviewer 2 Report

This interesting manuscript is generally well written. When corrected it will make an important contribution to the literature. That said there are a few “Chinglish” issues.

Line 44: change to read “…the key is to identify the viral pollution source…”

Line 56: I think you mean…“do not match rRNA…”

Iine 179: change to read “…the main interfering pathogen…”

Line 180: “this may be related to the match between” ….so I think you may want to say something to the effect that spurious priming may have occurred against this pathogen. It reads like they are supposed to match

Line 194: change to read “…essential for either tracing the contamination event or to identify…”

Line 202: change to read “…target sequences still represent a small proportion…”

Line 219: change to read “…or immune assays that have been previous identified but is not suitable for unknown pathogen metagenomics.”

Line224-225: I am unsure about the meaning of the sentence. Perhaps change to read “By excluding a large amount of metagenomic information, and only analyzing specific viruses, the probability of obtaining a whole genome of the target virus is improved, potentially reducing sequencing costs substantially.”
